# Non-Contact, Continuous Sampling of Porous Surfaces for the Detection of Particulate and Adsorbed Organic Contaminations by Low-Temperature Plasma Coupled to Ion Mobility Spectrometer

**DOI:** 10.3390/s23042253

**Published:** 2023-02-17

**Authors:** Izhar Ron, Hagay Sharabi, Amalia Zaltsman, Amir Leibman, Mordi Hotoveli, Alexander Pevzner, Shai Kendler

**Affiliations:** 1Department of Physical Chemistry, Israel Institute for Biological Research, Ness Ziona 74100, Israel; 2Department of Environmental, Water and Agricultural Engineering, Faculty of Civil & Environmental Engineering, Technion–Israel Institute of Technology, Haifa 32000, Israel; 3Department of Environmental Physics, Israel Institute for Biological Research, Ness Ziona 74100, Israel

**Keywords:** ion mobility spectrometry, on-site analysis, sampling

## Abstract

Chemical analysis of hazardous surface contaminations, such as hazardous substances, explosives or illicit drugs, is an essential task in security, environmental and safety applications. This task is mostly based on the collection of particles with swabs, followed by thermal desorption into a vapor analyzer, usually a detector based on ion mobility spectrometry (IMS). While this methodology is well established for several civil applications, such as border control, it is still not efficient enough for various conditions, as in sampling rough and porous surfaces. Additionally, the process of thermal desorption is energetically inefficient, requires bulky hardware and introduces device contamination memory effects. Low-temperature plasma (LTP) has been demonstrated as an ionization and desorption source for sample preparation-free analysis, mostly at the inlet of a mass spectrometer analyzer, and in rare cases in conjunction with an ion mobility spectrometer. Herein, we demonstrate, for the first time, the operation of a simple, low cost, home-built LTP apparatus for desorbing non-volatile analytes from various porous surfaces into the inlet of a handheld IMS vapor analyzer. We show ion mobility spectra that originate from operating the LTP jet on porous surfaces such as asphalt and shoes, contaminated with model amine-containing organic compounds. The spectra are in good correlation with spectra measured for thermally desorbed species. We verify through LC-MS analysis of the collected vapors that the sampled species are not fragmented, and can thus be identified by commercial IMS detectors.

## 1. Introduction

On-site chemical analysis is an essential task for first responders, such as law enforcement personnel, who need to examine surfaces suspected of being contaminated with adsorbed hazardous materials, such as explosives, illicit materials, chemical warfare agents or other accidentally released hazards. Such materials can be solid particles, or droplets, adsorbed deeply within a porous surface such as asphalt, sidewalks, building walls, vehicles, plants, or dissolved inside polymeric layers of different objects. Mapping an area after a chemical incident involving the release of liquid droplets or particulate matter requires high throughput sampling of various surfaces, many of which are corrugated.

Due to the complexity of sampling such surfaces, such mapping is performed mainly by sampling the vapors to a field portable detector, such as an ion mobility spectrometer (IMS), mass spectrometry (MS), and flame photometric detector (FPD) [1,2,3,4]. Numerous works demonstrated the use of such techniques combining desorption methods for the detection and mapping of explosives and drugs [5,6], toxic industrial compounds [7], volatile organic compounds (VOC) [8] and chemical warfare agents [9]. Sampling is based on collecting a minute amount of vapors released from the analyte or wiping an area of a few square centimeters with a swab, followed by thermal desorption of the collected sample. This practice has several drawbacks: mapping surface contamination is slow and not continuous, thus resulting in low spatial resolution. Additionally, the collection of large amounts of substances (both analytes and benign materials) will lead to contamination of the detection device and to memory effects; thermal desorption is energetically inefficient, thus reducing the operation times of the device. One of the most significant drawbacks of using these techniques is that collecting samples from corrugated surfaces using swabs may be impractical in many cases [10,11]; the alternative approach of sampling the vapor, as demonstrated by Kendler et al., requires a highly sensitive and complicated sensing setup and applies only to volatile and semi-volatile compounds. Additionally, large efforts are carried out for developing spatial sampling and analysis in order to improve autonomous and unmanned sensor deployment and accurately determine source distribution [12,13,14,15].

Hence, developing a sampling method that will allow the operator to perform fast and efficient sample introduction into a vapor analyzer without using solvents, disposables, or any sample preparation, may be valuable for many tasks.

During the last two decades, a novel analytical approach has been developed that deals with both the ionization and desorption of analytes in the open air for their analysis using, in most cases, mass spectrometry (MS) [16,17,18,19,20,21]. This field of research is termed ambient ionization, or ambient MS, and includes several different methods for ambient ionization, such as desorption electrospray ionization (DESI) [22] and direct analysis in real-time (DART) [23]. These techniques enable the application of high-performance analytical detection methods for complex situations in which analytes with low vapor pressure are sampled from realistic surfaces with no sample preparation, such as explosives and narcotics [24]. A promising technique for non-contact sample desorption and ionization is based on low-temperature plasma (LTP). Plasma is a general term describing a gas mixture of a charge-carrying particles such as ions and electrons. Non-thermal plasma such as LTP can be realized in a dielectric barrier discharge (DBD) configuration, where there exists at least one dielectric barrier and the spacing between the two electrodes is on the order of 0.1–10 mm [25]. Operation of a high voltage inside a carrier gas flow leads to the formation of a jet of ionized molecules, radicals, high-energy photons, and electrons with a kinetic energy of a few electron volts (eV) [26,27,28]. These reactive species lead to the dissociation and excitation of analytes in the surroundings. DBD offers several unique features: high dissociation ability at low temperatures, low energy consumption, simple and tunable configuration, operation in ambient pressure, and high durability. All these features have made this technique attractive for analytical devices. In addition, the low power consumption of only a few watts, with respect to the high electron density, is a promising feature for miniature, lab-on-a-chip type devices [29]. Specifically, certain configurations enable directing the jet outside the low-temperature plasma apparatus, thus allowing for various non-contact surface sampling schemes [30]. Once the LTP jet hits a surface, the energetic particles release adsorbed species in a process similar to chemical sputtering. Helium-based plasma introduces particles of approximately 20 eV, sufficient for ionizing many organic compounds through Penning ionization as well as atmospheric gases and water vapors, making it a potential ionization source for ion spectrometry [31]. The LTP jet in the ambient can interact non-destructively with various samples or objects, gaseous, liquid, solid, or aerosol, through desorption and soft ionization mechanisms. Realizing a sampling probe based on LTP is possible with straightforward means, and prototypes suitable for on-site operation have been demonstrated [32,33,34]. In addition, it has been studied for its ionization mechanisms and efficiency and was found to be a highly effective ion source compared to the more conventional electrospray ionization (ESI) and atmospheric pressure chemical ionization (APCI) sources, with potential advantages in the polarity range of molecules that can be ionized in this reaction [35,36]. Most studies demonstrated using LTP for sampling various objects and surfaces using MS as the analytical method [37,38,39,40,41,42,43,44,45,46]. However, ion mobility spectrometry (IMS) is a much more widely used method for the applications discussed in this work, namely on-site analysis of hazardous materials [47]. In IMS, ionized molecules travel along a drift tube under an electric field gradient and are separated according to their mobility in atmospheric pressure [48]. Several commercial-off-the-shelf (COTS) IMS instruments have been designed for on-site operation and offer a simple, handy, fast, and robust tool for real-time chemical analysis. While analysis is performed in the gas phase, several of these devices are equipped with built-in or add-on thermal desorption modules to allow also the analysis of liquids and powders collected with a sampling swab [49,50].

Extending the use of vapor analyzers for solid and liquid samples is achieved at the expanse of size and cost of the devices, limiting the field deployment. Thermal desorption devices also suffer from relatively high energy consumption and memory effects due to the introduction of bulk material into the device [51]. LTP has been demonstrated as an ionization and desorption source for IMS only on several occasions for laboratory IMS and its derivative differential ion mobility spectrometry (DMS) [52,53]. In these works, the ionization efficiency of various analytes has been investigated and compared to commonly used sources such as corona discharge and APCI sources of the radioactive type, i.e., ^63^Ni.

This study presents the use of a home-built LTP apparatus coupled to a commercial handheld IMS device working in vapor analysis mode. The LTP probe is used for a non-contact, non-thermal sampling unit for COTS IMS vapor analyzers. We successfully sample several non-volatile organic compounds from realistic surfaces such as asphalt and fabrics in a fast, continuous, non-contact, solvent-free mode of operation. The capability of the non-destructive sampling from highly corrugated surfaces using LTP is compared to the destructive thermal desorption.

## 2. Materials and Methods

### 2.1. Chemicals

Dodecylamine (>99%) and nicotinamide (98%) were obtained from Sigma-Aldrich (Ness-Ziona, Israel) and selected as amine-containing compounds, simulating non-volatile pharmaceutical substances.

### 2.2. LTP Apparatus

The LTP apparatus was built in-house based on configurations previously described in the literature. To date, LTP has been realized in two main configurations. In the coaxial or cross-field configuration, the LTP probe is built from a glass tube and a stainless-steel rod at its center that serves as ground. A copper foil wrapped outside the quartz tube serves as a high-voltage electrode [30]. Alternatively, the electric field can be co-aligned to the electrodes in a family of configurations that are termed linear field [54,55,56]. While this configuration may be even simpler to construct, it also showed more plasma activity and stability at the downstream region from the outlet of the probe [57]. A photo and scheme of our homebuilt LTP probe are shown in Figure 1. The apparatus was assembled using an empty quartz tube (i.d.—1.4 mm, o.d.—3 mm). Copper foil (3M shielding copper foil tape) was wrapped outside the tube at two locations. The wider electrode, closer to the outlet, serves as the high-voltage electrode, and the other serves as the ground. Gas (Helium 99.999%) is streamed through the tube controlled by a mass flow controller (AliMC 2slpm, Alicat Scientific, Tucson, AZ, USA). The LTP probe was connected to a PVM12—12 Volt plasma generator (Information Unlimited, Amherst, NH, USA), providing high voltages in the range of 1–15 kV at frequencies in the range of 20–50 kHz under a current limit of 3A.

### 2.3. IMS Analysis

Experiments were performed using a COTS handheld IMS detector, LCD 3.3 (Smiths Detection, Edgewood, MD, USA). The LCD 3.3 duty cycle is 5 s, which is considered, for this application, a continuous and real-time operation. Plasmagrams (ion current vs. drift time) are recorded throughout the measurement using TrimScan software (Smiths detection). The use of certain commercial equipment in this work does not imply recommendation or endorsement by IIBR, nor does it imply that the products identified are necessarily the best available for the purpose.

### 2.4. Thermal Desorption Reference Measurements

In reference experiments, a home-built thermal desorption (TD) module was used to desorb the analytes from the surfaces. The TD module was made of a glass cylinder wrapped with NiCr resistive heating wire (2 ohm/cm resistance, power supply-Zero 36 Lambda). The module was used to heat the sample to 250–270 °C for 20–40 s. A piece was taken off the sample and placed inside the TD module; this type of operation is destructive and slow hence appropriate for reference experiments and not intended for operational use.

### 2.5. Thermal Imaging of the LTP Jet

A thermal imager capable of measuring between −20–1200 °C, with a 240 × 320 pixels sensor and a thermal sensitivity of 0.05 °C (Ti400, Fluke, Everett, DC, USA), was used to image the LTP jet temperature.

### 2.6. Characterization of the Desorbed Vapors

In addition to the IMS measurement, the desorbed species’ chemical nature and the desorption rate were studied in a complement methodology. A sample was prepared by depositing a solution containing a model compound in methanol on glass and was left to dry for a few minutes. The sample was then placed in a glass vessel connected to an impinger filled with water as the collecting solvent, connected to a vacuum at a flow rate of 1 L/min. Vapors emanating from the sample upon operation of the LTP or the TD were trapped inside the collecting solvent that was later analyzed by LC-MS (Ultimate 3000—LC, LCQ Fleet—MS, Thermo Fisher Scientific, Waltham, MA, USA). Before the analysis, the LC-MS was calibrated using standard solutions of the model compounds at six different concentrations. LC solutions consisted of an aqueous solution with 0.1% formic acid (solution A) and a methanolic solution with 0.1% formic acid (solution B). The analyte concentration in the collecting solvent was determined also by measuring the light absorption in the ultraviolet and visible (UV-Vis) parts of the electromagnetic spectrum in the range 260–264 nm (Dionex, Ultimate 3000 diode array, Thermo Fisher). LC analysis was carried out under the following conditions: 150 × 2 mm column, Gemini C18 3 µm (P/N OOF-4439-BO, Phenomenex, Torrance, CA, USA).

## 3. Results and Discussion

### 3.1. LTP Operation and Characterization

Studying LTP as a source for the desorption of analytes from surfaces to the inlet of an IMS vapor analyzer was carried out through a series of sampling and detection experiments. The LTP apparatus was operated by applying several kV at the lower part of the frequency range of the power supply (~20 kHz). The He flow was stabilized at 1 L/min. Under these conditions, a narrow visible plasma jet was formed, protruding 3–4 cm from the outlet of the apparatus, Figure 2. Using a thermal imager, the temperature of this jet outside the apparatus seemed to be around room temperature, i.e., 23 °C, as seen in Figure 2C. The jet itself is not visible in the thermal image against the background. The surface temperature at the jet’s hit point was measured using a thin thermocouple. Figure 2A shows the relation between the current flowing through the power supply to the LTP probe and the temperature measured by the thermocouple. It can be seen that at the lowest current, the temperature of the surface at the hit point of the plasma jet is around 40 °C, and it goes up to 60 °C at 0.4 A. Higher temperatures can be obtained by increasing the current, but such conditions are beyond the scope of this work. It can be concluded from this part that a stable, few centimeters long LTP jet can be formed at a power consumption of just a few watts and that the interaction at the surface involves minor heating, far from the temperatures reached by thermal desorbers.

### 3.2. Characterization of Vaporized Products

Next, the content of the vapor emanating from the surface upon the operation of the LTP was analyzed using an LC-MS, as described earlier. Both model compounds (nicotinamide, NA, and dodecylamine, DA) contain two amine groups and give rise to an intense signal in the IMS. These two molecules are solid at room temperature, characterized by mild boiling points of 334 °C and 259 °C, respectively, typical of several drugs or other pharmaceutical substances. This experiment was designed to study the vaporized species’ identity and evaluate the adsorbed analytes’ desorption rate. The characterization of the chemical emitted during the operation of the LTP device was performed by comparing the IMS results using TD and validated by trapping the analytes in a liquid which was then analyzed using an LC-MS device. These two techniques complement each other as the TD-IMS is operated in real-time. The LC-MS, which is considerably slower, provides definite chemical identification and is a much more sensitive and selective device [58].

The experimental setup is shown in Figure 3A, and the LC-MS analysis of the nicotinamide (NA) sample is shown in Figure 3B. It can be seen that the main product in the collected vapor phase has an *m*/*z* of 122.94, which corresponds to a protonated NA (the molecular weight of NA is 122 g/mol). For the surface contaminated with dodecylamine (DA), the mass spectrum of the main fraction contained a dominant peak at *m*/*z* 186.13, which corresponds to a protonated DA molecule (the molecular weight of DA is 185 g/mol), as shown in Figure 3C. These observations support the assumption that desorption by LTP does not lead to the fragmentation of these substances and allows the transfer of intact analytes from surfaces to the gas phase.

During four minutes of operation, 6 µg of NA and 4.5 µg of DA were collected. According to this result, the evaporation rate of a sample deposited on a glass substrate subjected to the LTP jet was 1 µg/min. Such desorption rate is sufficient for rapid identification of deposits by an IMS vapor analyzer with an intrinsic sensitivity on the order of a few tens of nanograms.

### 3.3. LTP-IMS of Sample Deposited on Glass Surfaces

Since glass is a relatively inert surface, its interaction with the sample and other benign materials that the LTP may also vaporize is weak, making it a convenient starting point for this study. During the experiments, the LTP and the IMS were positioned a few centimeters from the surface (Figure 4A). The IMS spectra (plasmagrams) were recorded in real-time at five seconds duty cycle. In a control experiment in which the glass was clean, only the background reactant ion was seen, and no product ion was observed in the plasmagram. Similar results were obtained in a different control experiment with the contaminated glass substrate without applying the high voltage or when the LTP jet was directed to a non-contaminated part of the glass substrate. Operation of LTP on a glass surface contaminated with DA or NA molecules led to clear product ion peaks in the plasmagrams, as seen in Figure 4B,C, respectively. Similar results were obtained using destructive thermal desorption, which is applicable in the case of glass surfaces. Based on the knowledge gained from the LC-MS control experiments, the palsmagrams, shown in Figure 4, are assigned to the protonated sample molecules for both LTP and TD. The good correlation between the LTP-acquired and the TD-acquired spectra indicates that the LTP is a soft method for desorption and ionization which does not lead to fragmentation of the target molecule, hence allowing the detection device to easily identify the desorbed products.

### 3.4. LTP-IMS on Preserving Surfaces

A solution containing 50 µg of each model compound was drop cast on an asphalt surface and a suede shoe to study the potential of LTP to desorb analytes from realistic surfaces. As mentioned earlier, LTP sampling holds the potential to penetrate and interact with substances that were adsorbed inside substrates that can be considered preserving surfaces, from which sample pickup would not be efficient using the traditional means of wiping papers or swabs. In such samples, the analyte might be absorbed in the substrate’s pores or dissolved and diffuse to deeper parts of the substrate. In both cases, in contrast to flat and inert glass surfaces, the analyte is inaccessible to TD without intensive and destructive manipulations, which are unsuitable for real-time on-site analysis.

Figure 5A shows the obtained plasmagrams in the positive channel for sampling with an LTP probe from an asphalt surface contaminated with DA, compared to sampling under the same conditions from a glass surface. It can be seen that all three product ions of DA that appear in the contaminated glass sample also appear in the contaminated asphalt sample. There can be seen small shifts in the drift time of the peaks. These shifts can result from differences in water clustering around the ionized analyte, as LTP may reduce the number of available water molecules in the sampled area. Figure 5B shows the time trace of the dominant peak (drift time—8.54 ms) during a four minute operation of the LTP while scanning the surface by moving the LTP to different areas on the surface. This time trace shows that within a minute of the scan, the jet reached the contaminated area, and a signal was observed for prolonged periods of 10 s between several contaminated areas on the interrogated surface. This behavior differs from what is common when working with swab samples and thermal desorption, where a single spectrum represents the entire surface sampled with a swab. This example demonstrates the potential of LTP sampling to serve both as a non-contact and a continuous sampling method, both features presenting significant advantages over the current surface sampling mode of operation. The scan option is desirable for applications such as seeking local contaminations, for example, cargo sampling, where such a continuous spatial scan reduces the chance of false negative events originating from, e.g., areas inaccessible for conventional swabs (grooves, etc.). Additional tests for other surfaces were performed; Figure 5C,D shows the results obtained in the case of contaminated shoe surfaces. For both compounds, the product ion signals are acquired with good agreement with those acquired from a glass surface.

Due to the nature of the surface, analysis based on swabs is expected to have a low pickup yield, and direct heating is impractical. This is also evident from the plasmagram showing a higher ion signal with an added dimer, suggesting the efficient pickup from glass using the LTP method.

## 4. Conclusions

This study demonstrates the development and operation of a low-cost, simple construction LTP probe coupled to a commercial, handheld IMS vapor analyzer for the analysis of several surfaces contaminated with two amine-containing non-volatile compounds. In control experiments using LC-MS, it was shown that the vapors released from the surface upon LTP operation are the unfragmented analytes. The analytes signals appeared shortly after the operation of the LTP on the surfaces once the LTP hit a contaminated spot. IMS signals are highly correlated to those acquired for thermally desorbed analytes using a home-built thermal desorber apparatus. However, it is noted that the TD-based technique is used for comparison as it is slow and destructive hence it is less favorable for the application described here. Moreover, the pickup yield from such corrugated surfaces with a swab is expected to be inefficient.

LTP, on the other hand, was demonstrated for non-contact sampling, operated centimeters away from the interrogated surface. The LTP enables continuous scanning and sampling of the surface. These features result in a significant advantage in overcoming carryover and memory effects due to contamination of the analyzer with bulk materials. Additionally, LTP is less energetically demanding, shown to operate using only a few watts, and will allow for longer operation times. The option to scan the surface and follow the signal is also desired when trying to find localized contaminated spots or spots that are not accessible with a sampling swab.

According to the measured desorption rate, which was found to be around 1 µg/min., sampling of a few seconds may yield more than tens of nanograms of a vaporized analyte, which is within the sensitivity range of analyzers such as IMS. The known sensitivity of such detectors with sample introduction by thermal desorption of a swab is on the same order of tens of nanograms. Previous reports demonstrated a limit of detection of 150 ng of acetaminophen on an LTP-IMS laboratory setup [53], and to our knowledge the current study is the first demonstration of surface sampling using LTP and a commercial IMS hand-held device. The sensitivity of the demonstrated method implies that this method is suitable for trace detection, and optimizing LTP parameters may lead to even better sensitivities, such that the limiting factor would be the inherent detector sensitivity. We demonstrated LTP-IMS sampling and analysis from several types of contaminated surfaces, starting with non-preserving surfaces such as glass, then sampling from challenging, porous, and preserving surfaces such as asphalt and a leather shoe. In these experiments, the total amount of analyte was ~50 µg, and we demonstrated acquiring the analyets’ signals by LTP scanning the surface for several minutes. The current study showed high potential of this method for security and civilian applications of on-site, rapid non-contact screening and detection of objects suspected of being contaminated with non-volatile compounds such as explosives and illicit drugs at trace concentrations. We plan to further study and develop this method, and investigate the desorption process and parameters, the efficiency of sampling using other ionization gases, and the range and properties of analytes that can be effectively detected using this method. As we envision the use of the method, an LTP apparatus can be operated independently or attached to the detection device, as will be dictated by optimization of the geometry of the LTP apparatus surface detector. The parameters that will require optimization include the angle between LTP jet and the surface, the position of the detector with respect to the LTP device and the surface, the scan rate of the surface area and the distance from the surface. We have initiated an effort towards the realization of an LTP jet array that will allow the coverage of a larger area and will thus lead to improved sensitivities and operation times.

## Figures and Tables

**Figure 1 sensors-23-02253-f001:**
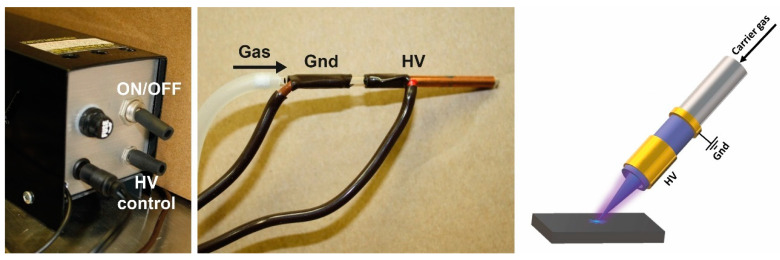
Plasma power source and home-built apparatus (**left**) and schematic description (**right**).

**Figure 2 sensors-23-02253-f002:**
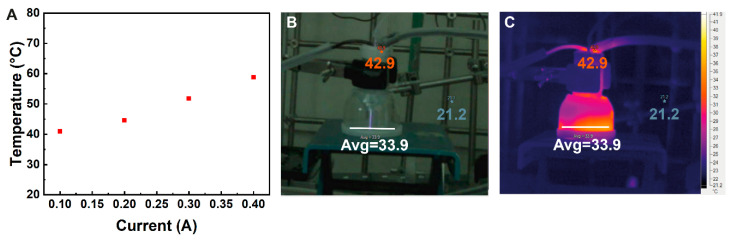
(**A**) Plasma jet contact point temperature on a glass surface as a function of plasma generator current (**B**) plasma jet contact with the surface and (**C**) the resulting average temperature at the surface as measured by a thermal imager.

**Figure 3 sensors-23-02253-f003:**
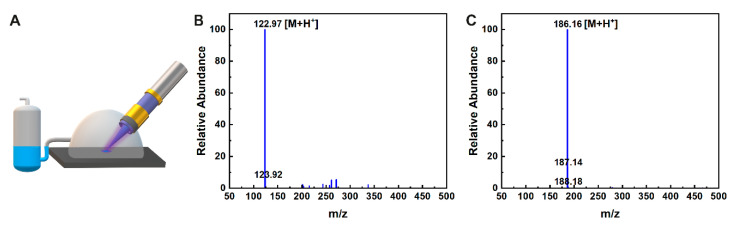
(**A**) Experimental setup for collecting vapor phase above a contaminated glass surface sampled with LTP probe. After sampling, the collecting-solvent is analyzed using an LC-MS. (**B**) LC-MS analysis of nicotinamide and of (**C**) dodecylamine.

**Figure 4 sensors-23-02253-f004:**
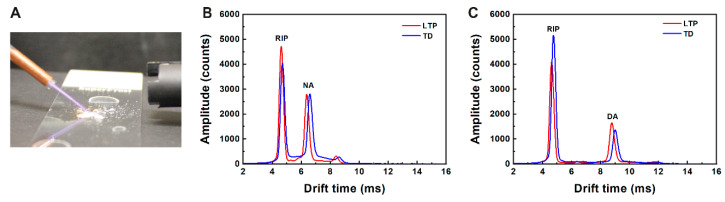
Experimental setup of sampling particulate contaminants from a glass surface into an IMS vapor detector using LTP (**A**) and ion mobility spectra (positive channel) of dodecylamine (**B**) and nicotinamide (**C**) sampled with LTP (red trace) compared to spectra obtained using destructive thermal desorption (blue trace).

**Figure 5 sensors-23-02253-f005:**
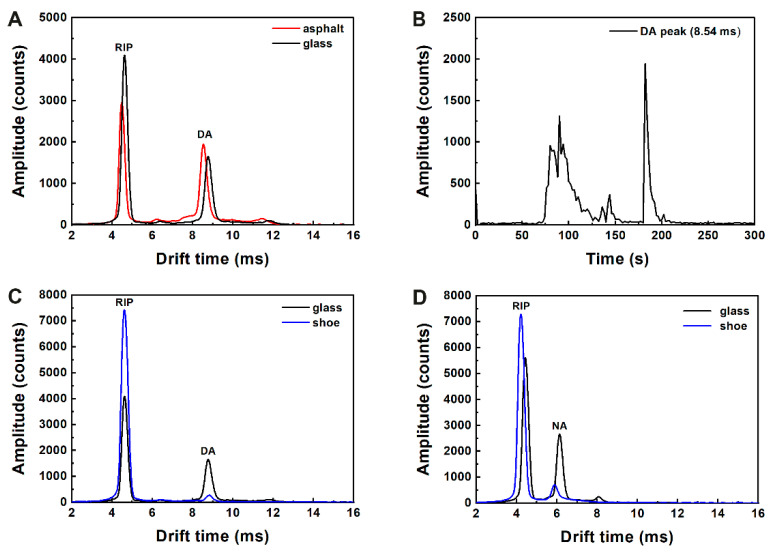
(**A**) LTP-IMS analysis of DA from glass and asphalt surfaces. (**B**) Time trace of the dominant IMS product ion peak of DA during a four-minute operation of the LTP. (**C**,**D**) LTP-IMS analysis of DA and NA from contaminated glass and shoe surfaces.

## Data Availability

Data in contained within the article.

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
