# Peer review of "Non-Contact, Continuous Sampling of Porous Surfaces for the Detection of Particulate and Adsorbed Organic Contaminations by Low-Temperature Plasma Coupled to Ion Mobility Spectrometer"

_sensors, 2023, doi:10.3390/s23042253_

Round 1

Reviewer 2 Report

In the review of the manuscript titled Non-contact, Continuous Sampling of Porous Surfaces for the Detection of Particulate and Adsorbed Organic Contaminations by Low-Temperature Plasma Coupled to Ion Mobility Spectrometer. The authors have provided a good description and the methodology is also fine. I would like to see this article publish but after some questions as follow;

1.     Why the authors just focused on organic contaminations?

2.      The authors claimed that they have demonstrated for the first time the operation of a simple, low cost, home built LTP apparatus for desorbing non-volatile analytes from various porous surfaces into the inlet of a handheld IMS vapor analyzer. They are requested to compare their results with the previously reported results.

3.     Why the spectra are in good correlation with spectra measured for thermally desorbed species?

4.     The ae authors are suggested to add some discussion about LC-MS analysis.

5.    The authors demonstrated Low-temperature plasma (LTP) as an ionization and desorption source for sample preparation-free analysis. They are requested to add a discussion to explain the procedure in detail.

Reviewer 3 Report

This study reported an operation of a low-cost, simple construction LTP probe with a commercial, handheld IMS vapor analyzer for the analysis of several porous surfaces contaminated with two amine-containing non-volatile compounds. The author verify through LC/MS anlysis of the collected vapors that the sampled species are not fragmented, and can thus be identified by commercial IMS detectors.

However,  some questions should be discussed in detailed:

1. The other parameters which affect the sensitivity should be explored, such as sampling angle, distance to the surface, the humidity....

2. Is the whole apparatus easily used in field detection? how to collect large surface samples without destruction?

3. The control experiment is a clean surface, however, the real sample surface is not clean, how to avoid matrix interference or false positives?

4. Does the angle of IMS to the sample surface affect the ion transport efficiency?  

Round 2

Reviewer 3 Report

The authors have revised the manuscript according to the reviewers' opinions. It can be accepted.